# Experimental Investigation of Electrochemical Capacitive Responses versus Pore Geometries through Artificial Nanotubes

**DOI:** 10.3390/mi14101909

**Published:** 2023-10-07

**Authors:** Jianyou Dai, Minghao Xu, Zhangshanhao Li, Shuoxiang Liu, Yuyao Wang, Lei Shan, Xiaohong Wang, Sixing Xu

**Affiliations:** 1School of Physics & Electronics, Hunan University, Changsha 410082, China; 2School of Integrated Circuits, Tsinghua University, Beijing 100084, China; 3College of Semiconductors (College of Integrated Circuits), Hunan University, Changsha 410082, China

**Keywords:** supercapacitors, porous electrode, electrochemical capacitance

## Abstract

Electrochemical supercapacitors have attracted significant attention due to their large capacity, high-power output, and long cycle life. However, despite extensive studies and advancements in developing highly porous electrode materials, little quantitative research on the impact of pore geometry on electrochemical responses has been conducted. This paper presents the first quantitative investigation of the relationship between electrochemical capacitive responses and pore geometries at the nanoscale. To achieve this, we constructed a uniform cylindrical pore array with controllable pore diameter and depth by using anodized aluminum oxide (AAO) to serve as a template and atomic layer deposition (ALD) technology for TiN conductive layer decoration. Our findings reveal that, at the nanoscale, increasing the specific surface area through pore diameter and depth does not proportionally increase the capacitive response, even at low scan rates. Meanwhile, we observe a critical pore parameter (170/5000 nm, diameter/depth), where the specific capacitance density and characteristic frequency dramatically decrease with a further increase in the pore aspect ratio. These results indicate that blindly pursuing the absolute specific surface area of the electrode material is not advisable. Instead, optimal pore geometry should be designed based on the desired operational conditions, and this work may serve as valuable guidance.

## 1. Introduction

Electrochemical supercapacitors possess high capacitance density, power density, and long cyclability, making them widely utilized for energy storage and power management [1,2,3,4,5,6,7,8]. Experimental evidence highlights the substantial impact of the nanostructures of supercapacitor electrodes on their electrochemical performance [9]. As a result, extensive research efforts have been dedicated to studying novel electrode materials with high porosity in recent decades [10,11,12,13,14,15]. However, a convincing and quantitative understanding of the influence of pore structure remains elusive. In other words, there is a lack of guidelines on how to optimize pore structure to enhance supercapacitor performance.

Several theoretical models have been proposed to explain the impact of pore topology [9,16,17,18]. For instance, De Levie has introduced the transition line model to provide a qualitative representation of the porous electrode electrochemical impedance versus frequency [19]; Murbach et al. and Pilon et al. theoretically examined the effects of the high-order nonlinear interfacial impedance of the pore structure [20]; and Huang et al. theoretically analyzed the ion diffusion resistance in porous electrodes under diverse boundary conditions [21]. Despite the abundance of theoretical investigations, only a few can be substantiated by compelling experimental evidence. This difficulty arises from the disorder present in practical porous electrode materials, which consist of a multitude of randomly distributed and interconnected pores. Consequently, it is challenging to manipulate experimental conditions (such as pore radius and depth) to validate theoretical models. To address this challenge, a promising approach is the development of an artificial pore array with a controllable structure as an experimental platform for quantitatively studying the impact of pore topology. As far as we were aware, the only study on this topic is reported by Peteghem et al., who achieved controllable pore structures on graphite surfaces through drilling technology and successfully measured the electrochemical responses [22]. However, due to challenges in microfabrication, the achieved pore sizes were on the millimeter scale, which is over four orders of magnitude larger than the pores found in typical electrochemical electrodes, disregarding some crucial nano-effects and thus lacking practical significance.

Herein, we present the first quantitative investigation into the correlation between electrochemical response and pore topology at the nanoscale. To achieve this, we developed an ordered nanopore array by anodizing an Al sheet to form AAO (Anodic Aluminum Oxide) with a controllable pore structure [23,24,25], followed by the atomic layer deposition of a conductive TiN layer [26,27,28,29], and the concept is illustrated in Figure 1. The AAO provides a controllable porous array with pore diameters ranging from 100 nm to 400 nm and pore depths ranging from 1 μm to 30 μm, serving as the electrode framework for supercapacitors. The TiN layer with highly controlled thickness not only enhances electrical conductivity but also exhibits favorable electrochemical performance. Our results are consistent with the theoretical prediction that an increase in surface area (by decreasing pore diameter or increasing pore depth) does not linearly translate to an increase in the capacitance of porous electrodes, even at low scan rates. Interestingly, we find that increasing the aspect ratio of the pores leads to a sharp decline in specific capacitance density and eigenfrequency for a critical pore parameter (170/5000 nm, diameter/depth). These findings suggest that pursuing the highest possible absolute specific surface area of the electrode material is not always the most appropriate approach. Instead, our study provides valuable insights for designing optimal pore geometries tailored to specific operating conditions.

## 2. Materials and Methods

### 2.1. Preparation of TiN/AAO Porous Array Electrodes

The preparation of AAO/TiN nanoporous array electrodes consists of two main steps: (i) anodizing of the aluminum foil to orderly grow porous alumina channels as the framework of the porous electrode and (ⅱ) deposition of TiN layer on the surface of ordered AAO template by ALD technique. The preparation of AAO involves two oxidation steps. After the first oxidation, the oxide layer is removed to leave behind pores with different spacing (different electrolytes are used for different pore spacings, with phosphoric acid used for a pore spacing of 450 nm). The second oxidation step is used to grow ordered channels, and the pore depth is proportional to the current multiplied by the time of use. Then, the TiN layer is deposited on the uniform AAO array using ALD (atomic layer deposition) technology to precisely control the pore size. The TiN layer with highly controlled thickness not only enhances electrical conductivity but also exhibits favorable electrochemical performance.

The TiN film is deposited by ALD (TFS-200, Beneq, Espoo, Finland) at 300 °C using TDMATi as precursors and NH_3_ as N source. The chamber pressure was controlled to approximately 0.3 mbar. The carrier gas was N_2,_ and the purge time was 8 s. The thickness of the TiN layer is controlled to be 15 nm after 110 cycles, which can improve electrical conductivity and electrochemical properties while not blocking the nanopores. The geometric parameters of the experimental samples for the proposed work are summarized in Table 1.

### 2.2. Characterization Method of the TiN/AAO Electrode

The TiN/AAO electrode samples were first characterized by Scanning Electron Microscope (SEM) and Transmission Electron Microscope (TEM). The SEM samples (surface and cross-section) were characterized by the MIRA3 LMH (TESCAN, Brno, Czech), in which the cross-section sample was prepared by cutting the TiN/AAO electrode with a hobby knife. The TEM sample was prepared by using Focused Ion Beam (FIB) from Zeiss-Auriga (Zeiss, Jena, Germany) (1 μm pore depth, 1 nA beam current with energy of 10 kV, and 2.5 nm beam spot), operated by Omiprobe-20 nanohand and observed by JEM-F200 (JEOL, Tokyo, Japan).

All cyclic voltammetry (CV) and electrochemical impedance spectrometry (EIS) results were measured from the symmetrical two-electrode system at the electrochemical workstation (CHI-660, Shanghai, China). Each electrode is controlled to be 1 cm × 1 cm and placed 1 cm away from the other. 1M TEMABF_4_/AN was applied as the electrolyte to prevent unwanted redox reactions. Due to the symmetry, the CV curve results can be considered as the series connection of two identical 1 cm × 1 cm electrodes. Therefore, the capacitance calculation formula for the three-electrode system of Equation (1) can be multiplied by 2 to obtain the capacitance of a single electrode. Thus, the direct current electrochemical capacitance can be calculated using Equation (2) as follows:(1)$$\begin{array}{c} C=\frac{s}{2mk\left(u_{1}-u_{2}\right)} \end{array}$$
(2)$$\begin{array}{c} C=\frac{s}{mk\left(u_{1}-u_{2}\right)} \end{array}$$
where C is the electrochemical capacitance, s is the CV integration area, k is the sweep speed, $u_{1}-u_{2}$ is the window voltage, and m is the sample mass or surface area.

Meanwhile, we tested the EIS curves at different pore diameters/pore depths and analyzed the variation of the AC capacitance with frequency by Equations (3) and (4) [30]
(3)$$\begin{array}{c} C_{\left(w\right)}^{}=\frac{-z_{\left(\omega \right)}^{\prime \prime }}{w{\left(z_{\left(w\right)}\right)}^{2}} \end{array}$$
(4)$$\begin{array}{c} C_{SS\left(w\right)}=\frac{c_{\left(w\right)}}{A} \end{array}$$
where $C_{SS\left(w\right)}$ is the specific surface area capacitance. By calculating the alternating current capacitance of a single electrode and distributing it evenly among the different parts of the electrode surface area, we can obtain the effective area at different frequencies for different geometric structures. $z_{\left(\omega \right)}^{\prime \prime }$ is the imaginary part of the impedance, $z_{\left(w\right)}$ is the magnitude of the impedance, w is the angular velocity, A=pi∗d∗L∗n is the surface area of the electrodes, and n is the number of holes.

### 2.3. Characterization of the TiN/AAO Porous Array Electrode

Figure 2a,b show SEM images of AAO of different pore diameters/depths after ALD deposition of the 15 nm TiN conductive layer. It can obviously be observed that these samples have relatively uniform pore diameters, flat pore depths, and similar pore spacing of around 460 nm. Owing to the anodic oxidation mechanism, the pore is close to circular structure in hexagonal frame. To verify the homogeneity of TiN on the pore surface, three samples were intercepted using FIB to prepare the top, middle, and bottom section samples for TEM characterization. As shown in Figure 2c, a relatively uniform 15 nm TiN layer on the AAO surface was clearly observed, while a high-resolution TEM (HRTEM) image is shown in Figure 2d.

## 3. Results and Discussion

The C-V curves and EIS measurements were conducted on electrodes with varying pore diameters and depths. The capacitance at both DC and AC frequencies was determined using Equations (1)–(4). It was observed that increasing the electrode surface area resulted in higher capacitance values. However, excessive growth of pore depths led to a decrease in specific capacitance per unit surface area. This is because the effective surface area of the electrode does not increase linearly with the overall surface area. Additionally, we found that when the ratio of pore diameter to depth exceeded 170/5000 nm, there was a significant decrease in the capacitance and frequency response of the electrodes.

### 3.1. Effect of Pore Depth

The experimental data in Figure 3a show the CV curves of the porous electrode with a fixed pore diameter (170 nm) and pore depths varying from 1/2/5/30 μm at a scan rate of 0.05 V/s. The capacitance values were calculated to be 109.48 μF/cm^2^, 216.07 μF/cm^2^, 365.13 μF/cm^2^, and 569.36 μF/cm^2^, respectively, by Equation (2). It can be seen that with the increase in the pore depth, the capacitance increased about five times, but the surface area increased by 30 times, indicating that the effective capacitance area did not increase linearly with the increase in pore depth, even at such a low scan rate.

Figure 3b summarizes the capacitance changes in porous electrodes with different pore depths as the scan rate varies from 0.005 V/s to 10 V/s. The capacitance shows a similar logarithmic decreasing trend as the scan rate increases. It can be observed that at a larger scan rate, the capacitance of an electrode with a smaller aspect ratio declines slower. The EIS results of those samples were plotted in Figure 3c. For the low-frequency part, the shallower the pore depth, the closer the curve is to the vertical lead line, indicating better capacitive performance. Excessive pore depth can make material transfer difficult. The semicircle structure at the high-frequency part (inset figure) indicates the diffusion impedance decreases with the enlargement of pore depth. This is because the increase in pore depth makes charge transfer more difficult. The AC capacitance density C_SS_ versus frequency is summarized in Figure 3d. It can be seen that the red curve (pore depth of 1 μm) is close to the yellow curve (pore depth of 2 μm) when the frequency is below 100 Hz, indicating that the pore geometry has a slight influence on the electrochemical response in low frequency and small aspect ratio. It can also be observed that when the pore depth increases to 30 μm, the C_SS_ decreases dramatically with the pore depth. Such a sharp decrease occurs because when the pore depth is larger than the ion penetration depth, the bottom of the pore cannot adsorb ions and form an electric double layer, resulting in a huge invalid surface area.

### 3.2. Effect of Pore Diameter

Figure 4a shows the C-V curves of the TiN/AAO electrode with a fixed pore depth (5 μm) and pore diameter varying from 370/270/170/70 nm at a scan rate of 0.05 V/s. The capacitance values were calculated to be 944.52 μF/cm^2^, 665.80 μF/cm^2^, 365.13 μF/cm^2^, and 119.44 μF/cm^2^, respectively, according to Equation (2). It can be seen that as the pore diameter decreases from 370 nm to 70 nm, the capacitance decreases nearly eight times while the surface area decreases about five times. Increasing the pore diameter increases the electrode surface area, resulting in higher electrode capacitance. However, the electrode capacitance is not linearly related to the electrode surface area. Smaller pore diameters may lead to capacitance values smaller than what is predicted by theory. Figure 4b summarizes the capacitance changes in TiN/AAO electrodes with different pore diameters as the scan rate varies from 0.005 V/s to 10 V/s. As the scan rate increases, the capacitance also shows a logarithmic decrease, and the rule that electrode capacitance does not linearly increase with surface area is applicable at different scan rates.

The EIS measurements of TiN/AAO electrodes with varying pore diameters are shown in Figure 4c. For the low-frequency part, the larger the pore diameter, the closer to the vertical plumb line the curve is, which means the better the capacitance performance and that a larger pore size makes material transfer easier. As the pore diameter becomes larger, the high-frequency semicircle becomes smaller, meaning the charge transfer impedance is smaller. The specific capacitance versus frequency is plotted in Figure 4d. It can be seen that the yellow curve (pore diameter of 270 nm) is approaching the red curve (pore diameter of 370 nm), indicating the similar effective capacitances of those two samples. This can be explained by the fact that the ion penetration depths are larger than 5 μm in the measuring frequency range so that the bottom part of the electrode pore structure can still be reached. Meanwhile, for the samples of pore diameter 170 nm and 70 nm, the specific capacitance is obviously smaller, especially when the frequency is higher than 10^3^ Hz.

## 4. Discussion and Conclusions

In this work, the effect of pore geometry parameters on the electrochemical performance of porous electrodes is discussed on a hundred-nanometer scale, but there are still some improvements that can be made. If the AAO substrate can be changed to a material with better conductivity or even superconductivity, it will be more in line with the De Levie model. At the same time, TiN is not a perfect bilayer material; there are still some interfacial reactions, and the diffusion impedance can be judged from the high-frequency semicircle of EIS. Meanwhile, if we can continue to increase the thickness of TiN and reduce the aperture to the ionic radius scale, according to reports, carbon-derived carbons with an average pore size ranging from 0.6 to 2.25 nanometers can be utilized [31]; the study investigated double-layer capacitance in organic electrolytes. The research results cast doubt on the long-standing axiom that pores smaller than the size of solvated electrolyte ions cannot facilitate charge storage.

In conclusion, this is the first time that the electrochemical properties of porous arrays with different geometrical parameters have been investigated at the 100 nm scale, as far as we know. The AAO porous arrays with uniform cylindrical structure/controllable size are the substrate, and the thin TiN layer is the conducting and electrochemical layer. The variation of capacitance with increasing pore depth/diameter is shown in detail, and we found that the capacitance does not increase linearly with the increase in the pore diameter and depth. When the aspect ratio reaches 30,000/170, the gain of the electrochemical effective adsorption area is deficient.

## Figures and Tables

**Figure 1 micromachines-14-01909-f001:**
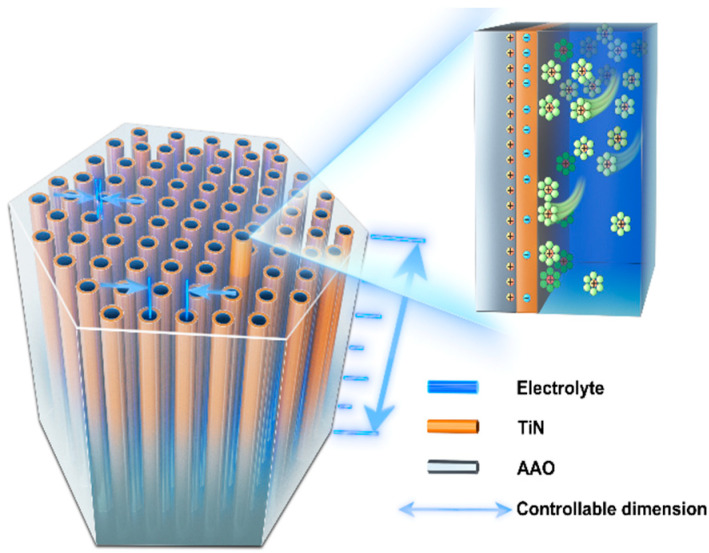
Structural schematic diagram of the TiN/AAO porous array electrodes.

**Figure 2 micromachines-14-01909-f002:**
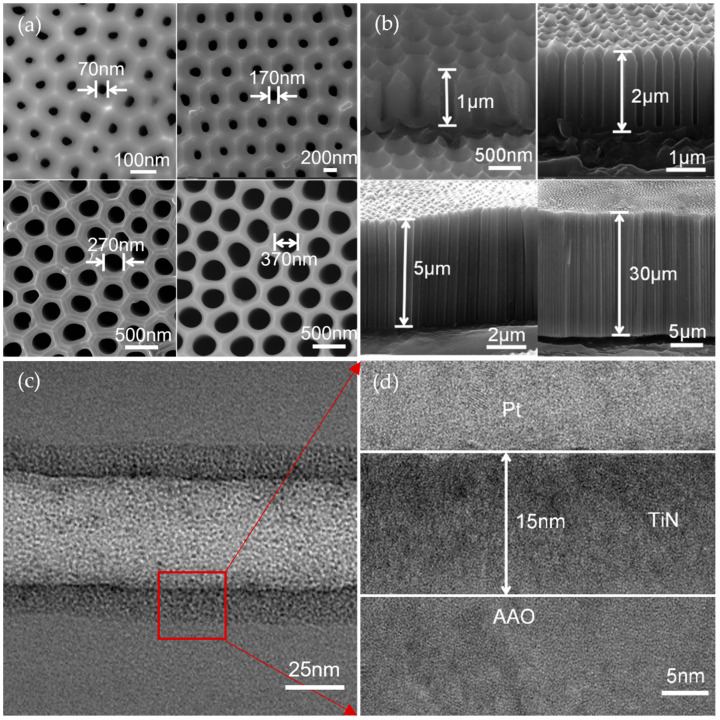
Characterization of the AAO/TiN Nanotube Array Structures: (**a**) SEM of the surface; (**b**)SEM of the cross-sections; (**c**) TEM of the cross-sections; (**d**) HRTEM of the cross-sections.

**Figure 3 micromachines-14-01909-f003:**
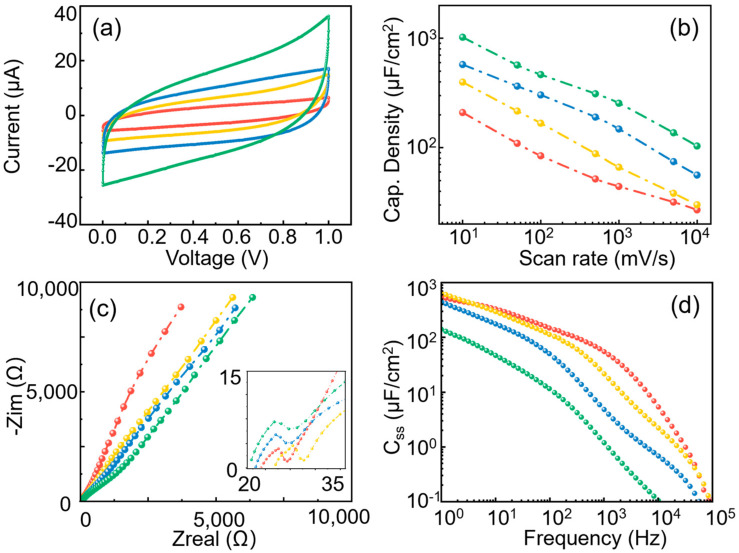
Comparison of different pore depths (red 1 μm; yellow 2 μm; blue 5 μm; and green 30 μm): (**a**) C-V curves at 0.05 V/s; (**b**) pore radii vs. specific capacitance; (**c**) Nyquist plot; and (**d**) specific surface area capacitance vs. rate.

**Figure 4 micromachines-14-01909-f004:**
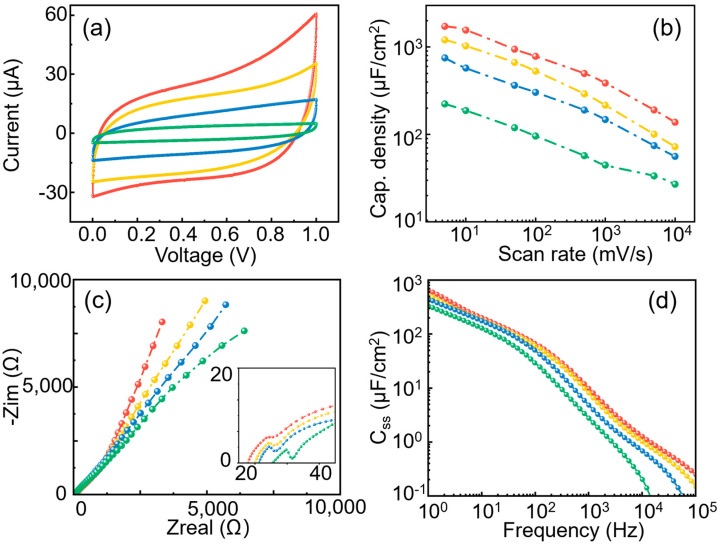
Comparison of different pore diameters (red—370 nm; yellow—270 nm; blue—170 nm; and green—70 nm): (**a**) C-V curves at 0.05 V/s; (**b**) pore diameter vs. specific capacitance; (**c**) Nyquist plot; and (**d**) specific surface area capacitance vs. rate.

**Table 1 micromachines-14-01909-t001:** Summary of the experimental samples with different geometric parameters.

Sample Parameter	Pore Diameter (nm)	Pore Depth (μm)
Varied pore depths (1M TEMABF_4_/AN)	170	1/2/5/30
Varied pore diameters (1M TEMABF_4_/AN)	70/170/270/370	5

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
