# Peer review of "Experimental Investigation of Electrochemical Capacitive Responses versus Pore Geometries through Artificial Nanotubes"

_micromachines, 2023, doi:10.3390/mi14101909_

Round 1

Reviewer 1 Report

This paper introduces an interesting porous structure with controllable nanoscale pore size and uniform cylindrical pore array, which is highly suitable for exploring and verifying electrochemical theories of porous electrodes. The effects of porous electrodes with different pore geometries on electrochemical properties are investigated, leading to practical conclusions and guiding suggestions. However, some minor issues should be considered before its publication:

1.       The analysis for figures 4a-b is too simplistic and there is a need for more detailed description and exploration.

2.       The mention of 1M Na2SO4 in Table 1 is inconsistent with the 1M TEMABF4/AN mentioned in the main text, please explain.

3.       At the end of the first paragraph in the Discussion & Conclusion section, it is mentioned that "if we can continue to increase the thickness of TiN and reduce the aperture to the ionic radius scale, more interesting phenomena will be seen". What phenomena does this refer to?

4.       Some important references about supercapacitors are missing in the Introduction part, such as:

1)              Origin and Regulation of Self-Discharge in MXene Supercapacitors. Adv. Funct. Mater., 2023, 16, 2208715.

2)              Unraveling and regulating self-discharge behavior of Ti3C2Tx MXene-based supercapacitors. ACS Nano, 2020, 14, 4916-4924. 

3)              Tailoring carbon nanomaterials via a molecular scissor. Nano Today, 2021, 36, 101033. 

4)              Additive engineering enables ionic-liquid electrolyte-based supercapacitors to deliver simultaneously high energy and power density. ACS Sustainable Chem. Eng., 2023, 11(14), 5685-5695. 

Reviewer 2 Report

This paper quantitatively investigates the relationship between electrochemical responses with porous electrode geometries at nanoscale, which is a very significant yet unsolved problem in the field of electrochemical supercapacitors. Meanwhile, the authors creatively fabricate the highly controllable TiN/AAO array as the experiment platform, making the results convincing. I suggest its publication in Micromachines after the consideration of following issues:

1.     Line 98-102, the symmetrical two-electrode testing and its capacitance calculation should be further elaborated to facilitate readers' understanding.

2.     Line 109, Equation (3) contains a variable defined by the authors. Please explain the meaning of this representation and why traditional areal capacitance density is not used.

3.     Line 174-175 and 150-152, the description of diffusion impedance is unclear.

4.     The detailed method about how to control the pore depth and pore size is missing.
